# Effect of Evolocumab on Vulnerable Coronary Plaques: A Serial Coronary Computed Tomography Angiography Study

**DOI:** 10.3390/jcm9103338

**Published:** 2020-10-18

**Authors:** Keiji Hirai, Shigeki Imamura, Aizan Hirai, Susumu Ookawara, Yoshiyuki Morishita

**Affiliations:** 1Division of Nephrology, First Department of Integrated Medicine, Saitama Medical Center, Jichi Medical University, 1-847 Amanuma-cho, Omiya-ku, Saitama-shi, Saitama-ken 330-8503, Japan; su-ooka@hb.tp1.jp (S.O.); ymori@jichi.ac.jp (Y.M.); 2Department of Internal Medicine, Chiba Cerebral and Cardiovascular Center, 575 Tsurumai, Ichihara-shi, Chiba-ken 290-0512, Japan; shigeimam@gmail.com (S.I.); aizanvaio4909@gmail.com (A.H.)

**Keywords:** coronary computed tomography angiography, coronary artery, vulnerable plaque, evolocumab, proprotein convertase subtilisin/kexin type 9 (PCSK9)

## Abstract

This study investigated the effects of evolocumab on vulnerable coronary plaques and factors associated with the change in stability and size of plaques in patients taking statins. Vulnerable coronary plaques were defined using coronary computed tomography (CT) angiography as having a density of <50 HU within the region of interest and a remodeling index ≥1.1. The changes in minimum CT density, remodeling index, and percent stenosis of vulnerable coronary plaques after six months of evolocumab administration were retrospectively analyzed in 136 vulnerable coronary plaques from 98 patients (68 men and 30 women; mean age: 72.9 ± 8.7 years) treated with a statin. The administration of evolocumab significantly increased the minimum CT density (39.1 ± 8.1 HU to 84.9 ± 31.4 HU, *p* < 0.001), reduced the remodeling index (1.29 ± 0.11 to 1.19 ± 0.10, *p* < 0.001), and decreased the percent stenosis (27.0 ± 10.4% to 21.2 ± 9.8%, *p* < 0.001). Multiple linear regression analysis revealed that baseline percent stenosis (standard coefficient (β) = −0.391, *p* = 0.002) independently correlated with the change in minimum CT density, whereas the baseline remodeling index (β = −0.368, *p* < 0.001) independently correlated with a change in the remodeling index. Evolocumab stabilized vulnerable coronary plaques and reduced their size. These results suggest that evolocumab protects against coronary artery disease progression in patients taking statins.

## 1. Introduction

Coronary artery disease is a major cause of death worldwide [1]. The majority of coronary artery disease cases can be attributed to the progression and/or rupture of vulnerable plaques [2]. Furthermore, a recent study has showed that the presence of vulnerable plaques and their progression are independent predictors for the development of acute coronary syndrome [3]. Therefore, the stabilization and regression of vulnerable coronary plaques are important for the prevention of acute coronary syndrome, myocardial infarction, and sudden cardiac death.

Coronary computed tomography (CT) angiography has recently emerged as a noninvasive imaging modality for the assessment of coronary plaques. It has high spatial resolution, permitting the visualization of a number of components of the vulnerable plaque. A previous study showed that coronary CT angiography, when compared with intravascular ultrasonography, had similar effectiveness at differentiating lipid-rich plaques from fibrous plaques [4]. In addition, coronary CT angiography was reported to be almost as effective as histopathologic analysis at detecting vulnerable coronary plaques [5]. Furthermore, the evaluation of coronary plaques using coronary CT angiography is a less invasive method and is repeatable and reliable.

A number of studies have shown that statins stabilize vulnerable coronary plaques and reduce their size in patients with coronary artery disease [6]. Therefore, statin therapy is considered the gold standard therapy for the treatment of patients with coronary artery disease [7]. However, coronary plaque progression is still observed in approximately 20% of patients undergoing intensive lipid-lowering therapy using statins [8].

Evolocumab is a fully human monoclonal antibody that targets proprotein convertase subtilisin/kexin type 9 (PCSK9) and lowers the circulating low-density lipoprotein (LDL)-cholesterol concentration by a further 60–70% in patients who are at high risk of cardiovascular events and who are undergoing statin therapy, but who require additional LDL-cholesterol reduction [9,10]. A recent randomized control trial showed that evolocumab reduced the coronary plaque volume in statin-treated patients [11]. However, it remains uncertain whether evolocumab can stabilize vulnerable coronary plaques or reduce their size. Furthermore, factors associated with the stabilization and regression of vulnerable coronary plaques under evolocumab treatment have not been determined. Therefore, in the present study, we investigated the effects of evolocumab on vulnerable coronary plaques and factors associated with changes in the stability and size of vulnerable coronary plaques in patients taking statins using serial coronary CT angiography.

## 2. Methods

### 2.1. Ethical Approval

This study protocol was approved by the institutional review board of Chiba Cerebral and Cardiovascular Center (J-480) and was performed in accordance with the ethical principles contained in the Declaration of Helsinki and its subsequent amendments. Informed consent was not required by the institutional review board because of the retrospective design of the study. Information regarding this study, including the right of patients to opt out, was uploaded on the website of Chiba Cerebral and Cardiovascular Center.

### 2.2. Patients

In our institute, most patients routinely receive carotid artery ultrasonography every 1–2 years to identify those at high risk of developing coronary artery disease [12]. Additionally, patients with carotid artery plaques undergo coronary CT angiography every 6–12 months to detect coronary plaques and/or to assess plaque vulnerability and size [12]. Intensive lipid-lowering therapy including statins and PCSK9 inhibitors are administered to these high-risk patients to prevent the development of acute coronary syndrome [12]. In Japan, PCSK9 inhibitors have been indicated for the treatment of patients with familial hypercholesterolemia or hypercholesterolemia who are at high risk for cardiovascular events and who do not adequately respond to statins. Therefore, evolocumab was administered to hypercholesterolemic patients with confirmed cardiovascular risk factors who did not achieve optimal LDL-cholesterol goals or who exhibited the progression of carotid intima–media thickness (IMT) despite statin therapy. We retrospectively analyzed the data from patients who had regularly visited Chiba Cerebral and Cardiovascular Center between April 2016 and March 2019.

The inclusion criteria were (i) age ≥20 years; (ii) the presence of one or more vulnerable coronary plaques defined by a CT density <50 HU within a region of interest and a remodeling index ≥1.1; (iii) treatment with evolocumab for ≥6 months: (iv) treatment with a statin for ≥12 months before the initiation of evolocumab; (v) coronary CT angiography performed before and 6 months after the initiation of evolocumab; and (vi) progression of carotid maximum IMT defined as carotid maximum IMT change >0.0 mm/year despite statin therapy, assessed using serial carotid artery ultrasonography before the administration of evolocumab. The exclusion criterion was renal replacement therapy. Patients who did not consent to evolocumab administration and who met the inclusion criteria but not the exclusion criterion, except for evolocumab administration, were included as the control group.

### 2.3. Study Design

This study was a single-center, retrospective comparative study. The study design is shown in Figure 1. Eighty-two patients included in the evolocumab group were compared with the control group of 16 patients. The baselines of each group were set as the day on which the initial coronary CT angiography was performed. Demographic and clinical data were obtained from the participants’ medical charts. Evolocumab was administered subcutaneously at a dose of 140 mg every 2 weeks on the same day of the week. Minimum CT density, remodeling index, percent stenosis for vulnerable coronary plaques, and lipid parameters, including circulating concentrations of total cholesterol, LDL-cholesterol, high-density lipoprotein (HDL)-cholesterol, triglyceride, and lipoprotein (a), were evaluated before and 6 months after the baseline in each group. Factors that were independently associated with the change in stability (evaluated by minimum CT density) and size (evaluated by remodeling index) of vulnerable coronary plaques during the administration of evolocumab were analyzed using multiple linear regression analysis.

### 2.4. Laboratory Methods

Blood and urine parameters were measured at the Department of Clinical Laboratory Medicine, Chiba Cerebral and Cardiovascular Center. Serum hemoglobin A1c (HbA1c) concentrations were given as National Glycohemoglobin Standardization Program format values. The estimated glomerular filtration rate (eGFR) was calculated by a modified equation of the Modification of Diet in Renal Disease study for Japanese patients with chronic kidney disease as follows: eGFR (mL/min/1.73 m^2^) = 194 × (serum creatinine)^−1.094^ × (age)^−0.287^ × (0.739 if female) [13]. Hypertension was defined as a mean systolic blood pressure ≥140 mmHg and/or mean diastolic blood pressure ≥90 mmHg, or current use of blood pressure-lowering drugs. Diabetes mellitus was defined as HbA1c level ≥6.5% or current use of blood glucose-lowering drugs or/and insulin treatment.

### 2.5. Measurement of Carotid IMT by Ultrasonography

Carotid IMT was determined using B-mode ultrasonographic imaging and a 7.5-MHz linear transducer (Aplio MX; Toshiba Medical Systems, Tokyo, Japan). Carotid IMT was determined as the distance between two echogenic lines corresponding to the lumen-intima interface and the media-adventitia interface on the far wall of the artery [14]. The carotid mean IMT was defined as the average of all mean IMT values obtained from left and right common carotid arteries, carotid bulb, and internal carotid arteries [15]. The carotid maximum IMT was defined as the greatest measurable carotid IMT on the left and right sides of the common carotid artery, bifurcation, and internal carotid artery [15]. All scans were carried out by experienced ultrasonographers. In this study, we considered worsening of the maximum IMT to represent the progression of coronary artery plaques, because the carotid maximum IMT was reported to correlate with the percentage of the coronary artery covered by plaque [16] and IMT worsening was associated with the risk of future coronary events [17,18]. The progression of carotid maximum IMT was defined as the carotid maximum IMT change >0.0 mm/year as previously reported [18].

### 2.6. Assessment of Coronary Plaques Using Coronary CT Angiography

Coronary plaques were assessed by coronary CT angiography using a 320-row multidetector CT (Aquilion One; Toshiba Medical Systems), 0.5 mm detector collimation, 350 ms gantry rotation time, 175 ms temporal resolution (with half-scan reconstruction), and up to 16 cm of coverage in the Z direction. The same scanning protocol was used to reduce the variation in the CT density of the plaque because the CT density of the lumen influences the CT density of the plaque [19]. For contrast-enhanced scans, contrast medium (Iopamidol, 370 mg/mL Iopamiron; Bayer Yakuhin Ltd., Osaka, Japan) was injected at a rate of 24.5 mg/kg/s for 12 s through a 20-G cannula placed in the right antecubital vein, followed by 20 mL of saline using a dual-syringe injector (Nemoto Kyorindo, Co., Ltd., Dual Shot GX, Tokyo, Japan). The scan was started approximately 7 s after the contrast medium reached the left ventricle. CT images were obtained during one to three heartbeats, depending on participants’ heart rate. All the scans were performed during a single breath-hold. Participants were administered metoprolol orally at a dose of 20 mg, 2 h before the scheduled scan if their heart rate was >65 beats/minute (bpm), and landiolol intravenously at a dose of 0.125 mg/kg if a heart rate of >65 bpm persisted. The raw CT scan data were reconstructed using algorithms optimized for retrospective electrocardiogram-gated segment reconstruction. The CT perfusion imaging data were analyzed using the Ziostation 2 visualization system (Ziosoft, Tokyo, Japan).

Coronary CT angiography images were analyzed by a single reader with 5 years of experience in coronary CT angiography imaging. A coronary plaque was defined as a clearly discernible structure larger than 1 mm^2^, within or adjacent to the vessel wall, which was clearly distinguishable from the vessel lumen. Tissue with a CT density below 0 HU was considered pericardial fat surrounding the vessel and was excluded from the analysis [5]. The CT density of the plaques was measured using several circular regions of interest (with areas of 1 mm^2^) in each plaque (Figure 2A) [20], and the minimum CT density of the plaque was determined. A low-attenuation plaque was defined as a plaque with minimum CT density < 50 HU. Previous studies defined a low-attenuation plaque as a plaque with minimum CT density < 30 HU [20,21]. Cademartiri et al. showed that the CT density of the plaque increased in accordance with increasing CT density of the lumen [19]. The CT density of the lumen was higher in our study than that in a previous study [21] (436 HU vs. 258 HU). Therefore, we set the cut-off value for low-attenuation plaques at 50 HU based on previous reports [22,23]. The remodeling index was calculated by dividing the maximal vessel diameter in the lesion by the mean of the proximal and distal reference vessel diameters (Figure 2B). Positive remodeling was defined by a remodeling index ≥ 1.1 [20,21]. A vulnerable plaque was defined as a plaque with low attenuation and positive remodeling [20,21]. The percent stenosis was calculated as a ratio of the minimal lumen diameter and the mean of the proximal and distal reference vessel diameters (Figure 2C) [24]. The CT density of the lumen was measured using a circular region of interest placed in the center of the lumen (Figure 2D). Our workstation did not have automatic coronary plaque analysis software; therefore, we could not investigate all coronary plaques in the whole coronary tree or assess plaque volume in the coronary arteries.

### 2.7. Statistics

Statistical analyses were carried out using JMP 11 software (SAS Institute Inc., Cary, NC, USA). Continuous variables were presented as means ± standard deviations for a normal distribution and as medians [interquartile ranges] for a non-normal distribution. Categorical variables were presented as numbers and percentages. Comparisons of laboratory parameters before and 6 months after the baseline in each group were performed using the paired *t*-test for normally distributed data and the Wilcoxon signed-rank test for non-normally distributed data. Comparisons of component ratios between the two groups were performed using Fisher’s exact test. Comparisons of clinical parameters between the two groups were performed using the Student’s *t*-test for normally distributed data and the Mann–Whitney *U*-test for non-normally distributed data. Parameters that appeared to be significantly correlated with the change in stability (evaluated by minimum CT density) and size (evaluated by remodeling index) of vulnerable coronary plaques during the administration of evolocumab in simple linear regression analyses (*p* < 0.10) were included in the multiple linear regression analysis to identify those that were independently correlated with the change in stability and size of vulnerable coronary plaques during the administration of evolocumab. For all tests, *p*-values < 0.05 were considered statistically significant.

## 3. Results

### 3.1. Patient Characteristics

Overall, 148 patients with vulnerable coronary plaques who underwent serial coronary CT angiography were identified; 124 of these patients were treated with evolocumab, and 24 were not. Forty-one patients treated with evolocumab did not meet the inclusion criteria and one patient met the exclusion criterion, resulting in the inclusion of 82 patients in the evolocumab group. Sixteen patients not treated with evolocumab were assigned to the control group (Figure 3). Thus, 136 vulnerable coronary plaques from 98 patients (68 men and 30 women; mean age: 72.9 ± 8.7 years) were analyzed. The baseline characteristics of the patients as well as the medications used in the two groups are shown in Table 1. There were no significant differences in clinical parameters between the two groups except for the proportion of patients taking eicosapentaenoic acid and the mean IMT of the carotid artery. All the participants were taking statins and six patients (6.1%) had a history of myocardial infarction. Among six patients, four (4.1%) underwent percutaneous coronary intervention. Two patients (2.0%) had severe carotid artery stenosis and they underwent carotid artery stenting. The percentages of the participants with diabetes mellitus, hypertension, and familial hypercholesterolemia were 76.5%, 50.0%, and 0.0%, respectively. Reasons to start evolocumab were unable to achieve LDL-cholesterol goals with the maximum tolerated dose of a statin in 41 patients (50.0%), and progression of carotid maximum IMT despite statin therapy in 41 patients (50.0%). The doses of each statin administered in the two groups are summarized in Table 2. All patients were asymptomatic and had no signs or symptoms of coronary artery disease, such as chest pain, chest discomfort, or shortness of breath. None of the patients developed acute coronary artery syndrome, including unstable angina, or acute myocardial infarction, during the study period.

### 3.2. Effects of Evolocumab on Vulnerable Coronary Plaques, Assessed Using CT Angiography

The stability (evaluated by minimum CT density) of the vulnerable coronary plaques in the evolocumab group was significantly increased from 39.1 ± 8.1 HU at baseline to 84.9 ± 31.4 HU after six months (*p* < 0.001) (Figure 4A). There was no significant difference in the CT density of the coronary artery lumen between baseline and six months in the evolocumab group (419 ± 62 HU vs. 418 ± 64 HU, *p* = 0.77) (Figure 4B). The size (evaluated by remodeling index) of the vulnerable coronary plaques in the evolocumab group was significantly decreased from 1.29 ± 0.11 at baseline to 1.19 ± 0.10 after six months (*p* < 0.001) (Figure 4C). The percent stenosis at vulnerable coronary plaque sites in the evolocumab group was significantly decreased from 27.0 ± 10.4% at baseline to 21.2 ± 9.8% after six months (*p* < 0.001) (Figure 4D). All CT angiography parameters (minimum CT density, CT density of the coronary artery lumen, remodeling index, and percent stenosis) after six months in the control group did not significantly change compared with the baseline (Figure 4A–D).

### 3.3. Factors Associated with the Change in Stability and Size of Vulnerable Coronary Plaques

Simple linear regression analyses revealed that the change in the stability of vulnerable coronary plaques significantly correlated with a history of myocardial infarction, baseline stability of vulnerable coronary plaques, and percent stenosis at vulnerable coronary plaque sites (Table 3), whereas the change in the size of vulnerable coronary plaques significantly correlated with change in LDL-cholesterol, LDL-cholesterol < 70 mg/dL, HbA1c, change in HbA1c, renin–angiotensin system inhibitor use, and baseline size of coronary vulnerable plaques (Table 4). We performed multiple linear regression analyses using variables that were marginally or statistically significantly correlated (*p* < 0.10) with a change in stability and size of vulnerable coronary plaques in the simple linear regression analyses. These analyses revealed that percent stenosis at vulnerable coronary plaque sites (standard coefficient (β) = −0.391, *p* = 0.002) independently correlated with a change in plaque stability, whereas baseline plaque size (β = −0.368, *p* < 0.001) independently correlated with a change in plaque size.

### 3.4. Changes in Lipid Parameters

The circulating concentrations of total cholesterol, LDL-cholesterol, triglyceride, and lipoprotein (a) in the evolocumab group were significantly decreased from 149.9 ± 28.5 mg/dL, 70.4 ± 21.5 mg/dL, 111 (81–156) mg/dL, and 14 (6–26) mg/dL at baseline to 92.5 ± 22.4 mg/dL, 19.3 ± 16.0 mg/dL, 84 (62–120) mg/dL, and 5 (2–14) mg/dL after six months, respectively (each *p* < 0.001). In contrast, the HDL-cholesterol concentration in the evolocumab group was significantly increased from 53.2 ± 12.6 mg/dL at baseline to 55.7 ± 13.3 mg/dL after 6 months (*p* < 0.001) (Table 5). The total cholesterol concentration in the control group was significantly decreased from 174.3 ± 38.3 mg/dL to 145.9 ± 22.6 mg/dL after six months (*p* = 0.021). The LDL-cholesterol, triglyceride, lipoprotein (a), and HDL-cholesterol concentrations after 6 months in the control group did not significantly change compared with the baseline (Table 5).

### 3.5. Changes in Other Laboratory Parameters and Adverse Effects

There were no changes in the other measured clinical and laboratory parameters (uric acid, HbA1c, eGFR, urine albumin/creatinine ratio, alanine aminotransferase activity, and creatine phosphokinase activity) between baseline and six months in the evolocumab or control groups (Table 5). Adverse effects were observed in one patient (leg cramps) in the evolocumab group. However, the patient was otherwise tolerant of evolocumab and its administration was continued.

## 4. Discussion

In the present study, we investigated the effects of evolocumab on vulnerable coronary plaques assessed using CT angiography and the factors associated with the change in stability (evaluated by minimum CT density) and size (evaluated by remodeling index) of vulnerable coronary plaques. We found that evolocumab increased the stability, reduced the size of vulnerable coronary plaques, and decreased the percent stenosis at vulnerable coronary plaque sites in patients taking statins. We also found that percent stenosis at vulnerable coronary plaque sites was negatively correlated with plaque stability, whereas baseline plaque size was negatively correlated with the change in plaque size. However, the stability and size of vulnerable coronary plaques did not change in statin-treated patients who did not receive evolocumab administration.

Atherosclerotic plaques are classified into two types: vulnerable plaques and stable plaques [25]. A vulnerable plaque is defined as a plaque rich in lipids, with a necrotic core, that is covered by a thin fibrotic layer, whereas stable plaques have a relatively thick fibrous cap and a smaller lipid core [25]. The amount of plaque with a low CT density positively correlates with the percentage of fatty tissue, determined using intravascular ultrasonography [26]. During the initial stages of atherosclerosis, coronary plaques are enlarged, which is accompanied by an increase in vessel diameter: a process called positive remodeling [2]. The percent stenosis, which is a ratio of the minimal lumen diameter at the site of the plaque to the mean of proximal and distal reference vessel diameters, did not correlate with coronary plaque area in coronary atherosclerotic lesions [27]. In contrast, the remodeling index, which is the ratio of the vessel diameter at the site of the plaque divided by the mean of the proximal and distal reference vessel diameters, strongly correlated with coronary plaque size in coronary atherosclerotic lesions [28,29]. In the present study, evolocumab increased the CT density and reduced the remodeling index of vulnerable coronary plaques. These results suggest that evolocumab stabilizes vulnerable coronary plaques and reduces their size in patients undergoing statin therapy. Furthermore, the baseline plaque size was negatively correlated with a change in plaque size. A previous study reported that the baseline coronary atherosclerotic plaque volume was negatively correlated with a change in coronary atherosclerotic plaque volume in patients undergoing statin therapy [8]. These findings suggest that evolocumab might be more beneficial in patients with increased vulnerable coronary plaque size. However, percent stenosis was negatively correlated with a change in the stability of vulnerable coronary plaques. It is thought that disturbed blood flow and endothelial shear stress caused by increased luminal stenosis are responsible for the progression and instability of atherosclerotic plaques [30]. These findings suggest that evolocumab might be less beneficial in patients who have increased percent stenosis at vulnerable coronary plaques site. Further studies are needed to investigate the factors associated with the plaque stabilization and size reducing effects of evolocumab on vulnerable coronary plaques in patients undergoing statin therapy.

Statins were reported to stabilize vulnerable coronary plaques and reduce their size assessed by coronary CT angiography [31,32]. A prospective observational study showed that rosuvastatin increased the minimum CT density from 7.8 HU to 33.8 HU and reduced the remodeling index of vulnerable coronary plaques from 1.16 to 1.06 over six months in patients with acute coronary syndrome [32]. In the present study, evolocumab increased the minimum CT density from 39.1 HU to 84.9 HU and reduced the remodeling index of vulnerable coronary plaques from 1.29 to 1.19 over six months in patients taking statins. These results suggest that evolocumab has plaque stabilizing and reducing effects on vulnerable coronary plaques in patients taking statins. Further studies are needed to confirm the efficacy of evolocumab on vulnerable coronary plaques in patients undergoing statin therapy using CT angiography. In our study, half of the patients exhibited carotid maximum IMT progression despite a serum LDL-cholesterol level < 70 mg/dL under statin therapy, and they were treated with evolocumab. The American College of Endocrinology Medical Guideline and guidelines of The American Association of Clinical Endocrinologists recommend achieving serum LDL-cholesterol levels < 70 mg/dL in dyslipidemic patients with coronary artery disease or in those with diabetes mellitus or chronic kidney disease stage 3 or 4 with one or more risk factors for atherosclerotic cardiovascular disease [7]. Our therapeutic approach might be beyond the currently used clinical guidelines. Recently, the early detection of subclinical atherosclerosis using non-invasive imaging modalities such as carotid ultrasonography and coronary CT angiography was proposed [12]. These modalities are considered useful for determining the need for more aggressive atherosclerotic cardiovascular disease preventive strategies [7]. In addition, intensive lipid-lowering therapy was suggested to halt plaque progression to prevent the first coronary event rather than waiting for the first coronary event [12]. Based on this evidence, evolocumab was administrated to patients with progressive subclinical atherosclerosis despite statin therapy to prevent the development of cardiovascular disease. Further prospective studies are necessary to determine the indication and efficacy of intensive lipid-lowering therapy including evolocumab in patients undergoing statin therapy.

Atherosclerosis is a chronic inflammatory disease in which macrophage activation and lipid loading play critical roles [33]. Injury to the vascular endothelium by multiple mechanisms, including hyperglycemia, hypertension, and dyslipidemia, induces the infiltration and retention of monocytes in the subendothelial space. Subsequently, these monocytes differentiate into macrophages, take up oxidized LDL-cholesterol via macrophage scavenger receptors, and are ultimately transformed into foam cells. This process leads to intimal thickening and the formation of a lipid core that contains lipid-laden foam cells, which is surrounded by a layer of connective tissue [33]. PCSK9 is a serine protease produced by hepatocytes, endothelial cells, vascular smooth muscle cells, and macrophages [34] that increases the clearance of LDL-cholesterol receptors by hepatocytes by facilitating the lysosomal degradation of the receptors, which leads to an increase in the serum LDL-cholesterol concentration [35]. Several studies showed that evolocumab reduced the LDL-cholesterol concentration by 60–70%, decreased triglyceride and lipoprotein (a) concentrations, and increased HDL-cholesterol [9,10]. The findings of these previous reports are consistent with the results of the present study. PCSK9 also inhibited ATP-binding cassette transporter A1 expression in macrophages by depleting LDL receptors, which reduced cholesterol efflux from lipid-laden foam cells [36]. Evolocumab was reported to facilitate the uptake of serum LDL-cholesterol by hepatocytes by inhibiting LDL-cholesterol receptor clearance [37], and promote ATP-binding cassette transporter A1 expression in macrophages by upregulating LDL receptor expression [36]. These findings suggest that evolocumab shrinks atherosclerotic plaques by reducing serum LDL-cholesterol concentrations and increasing cholesterol efflux from lipid-laden foam cells. However, in the present study, there was no correlation between LDL-cholesterol, change in LDL-cholesterol concentration, or LDL-cholesterol <70 mg/dL with a change in the stability and size of vulnerable coronary plaques during the administration of evolocumab. No associations were reported between serum LDL-cholesterol, change in LDL-cholesterol, and change in coronary atherosclerotic plaque volume in patients who achieved LDL-cholesterol concentration <70 mg/dl under statin therapy [8]. In the present study, the mean LDL-cholesterol concentration at baseline was well controlled by statin therapy (70.4 ± 21.5 mg/dL). It has been suggested that evolocumab has pleiotropic anti-atherosclerotic effects by decreasing oxidative stress, suppressing inflammatory cytokine production, and inhibiting macrophage accumulation [34]. These findings suggest that evolocumab stabilizes vulnerable coronary plaques and reduces their size by mechanisms independent of its LDL-cholesterol lowering effect in patients whose LDL-cholesterol levels are well controlled with statin therapy. Further basic and clinical studies are required to clarify the factors mediating the effect of evolocumab on vulnerable coronary plaques in patients undergoing statin therapy. In addition, large-scale studies are needed to investigate whether evolocumab prevents the development of coronary artery disease in statin-treated patients with vulnerable coronary plaques.

This study had several limitations. First, it was a single-center, retrospective, observational study, which may have been subject to reporting and patient selection biases. Second, the manual measurement of CT density might have been associated with greater intra-observer variability, which might have affected the results of the study. Therefore, further prospective, multicenter, randomized studies that incorporate an appropriate control group and the use automated coronary atherosclerotic plaque analysis software, are required to definitively evaluate the effects of evolocumab on vulnerable plaques in coronary arteries using CT angiography.

In conclusion, evolocumab stabilizes vulnerable coronary plaques and reduces their size in patients whose LDL-cholesterol levels are well controlled with statin beyond its LDL-cholesterol lowering effect, without inducing any serious adverse effects.

## Figures and Tables

**Figure 1 jcm-09-03338-f001:**
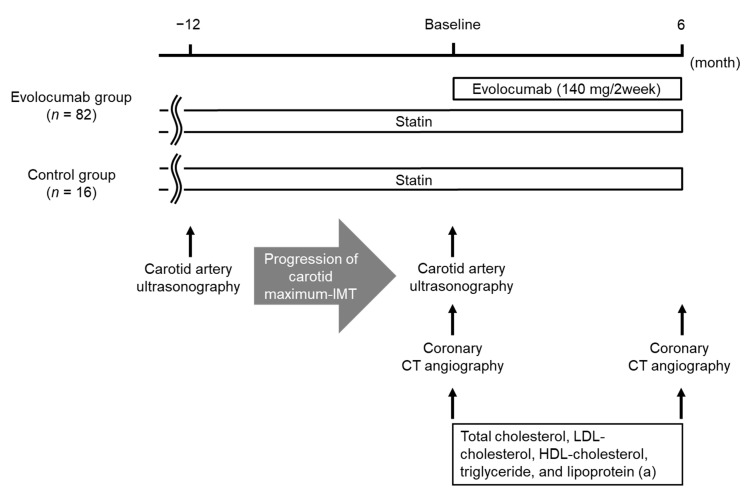
Study design. Abbreviations: CT, computed tomography; HDL, high-density lipoprotein; IMT, intima–media thickness; LDL, low-density lipoprotein.

**Figure 2 jcm-09-03338-f002:**
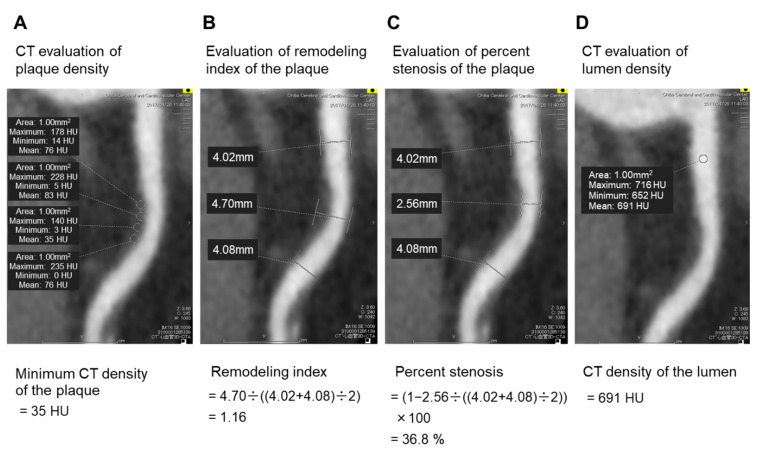
CT angiography imaging of the coronary artery. Several circular regions of interest (with areas of 1 mm^2^) were placed in the coronary plaque (**A**). Tissue with a CT density < 0 HU was considered pericardial fat surrounding the vessel and excluded from the analysis. The minimum CT density of the plaque was determined by comparing the mean CT density of each circular region of interest. In this case, the mean CT density of each circular region of interest was 76, 83, 35, and 76 HU; therefore, the minimum CT density of the plaque was 35 HU (**A**). The remodeling index was calculated by dividing the maximum vessel diameter in the lesion by the mean of the proximal and distal reference vessel diameters (**B**). In this case, the maximum vessel diameter in the lesion, the proximal reference vessel diameter, and the distal reference vessel diameter were 4.70, 4.02, and 4.08, respectively; therefore, the remodeling index was 1.16 (**B**). The percent stenosis was calculated as a ratio of the minimal lumen diameter to the mean of proximal and distal reference vessel diameters (**C**). In this case, the minimal lumen diameter, the proximal reference vessel diameter, and the distal reference vessel diameters were 2.56, 4.02, and 4.08, respectively; therefore, the percent stenosis was 36.8%. The CT density of the lumen was measured using a circular region of interest placed in the center of the lumen (**D**). In this case, the mean CT density of the circular region of interest was 691 HU; therefore, the CT density of the lumen was 691 HU (**D**). Abbreviation: CT, computed tomography.

**Figure 3 jcm-09-03338-f003:**
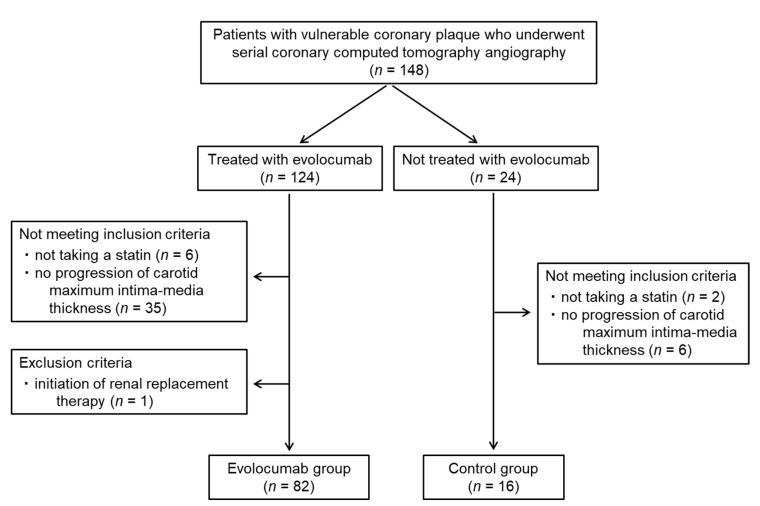
Patient flow diagram.

**Figure 4 jcm-09-03338-f004:**
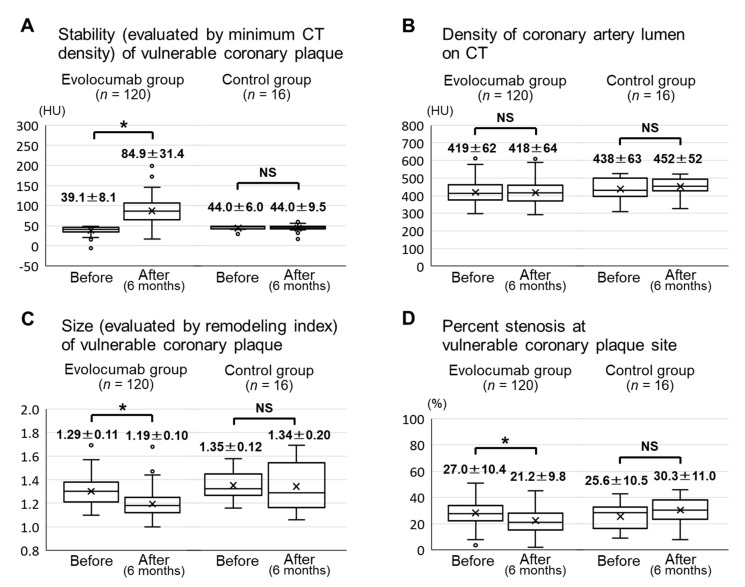
CT angiography parameters before and six months after the baseline in the evolocumab and control groups. Minimum CT densities of the vulnerable coronary plaques (**A**). CT densities of the coronary artery lumen (**B**). Remodeling indexes of the vulnerable coronary plaques (**C**). Percent stenosis at vulnerable coronary plaque sites (**D**). Abbreviations: CT, computed tomography; NS, not significant. *****
*p* < 0.001.

**Table 1 jcm-09-03338-t001:** Participant characteristics and medications at baseline.

		Evolocumab Group (*n* = 82)	Control Group (*n* = 16)	*p*-Value
Age (years)		73.4 ± 8.3	70.4 ± 10.2	0.23
Male sex (number, %)		56 (68.3)	12 (75.0)	0.77
Body mass index (kg/m^2^)		24.8 ± 3.6	23.8 ± 3.4	0.32
Systolic blood pressure (mmHg)		134.1 ± 15.3	135.1 ± 10.5	0.79
Diastolic blood pressure (mmHg)		78.7 ± 9.1	80.8 ± 8.6	0.40
Hypertension (number, %)		64 (78.0)	11 (68.8)	0.52
Diabetes mellitus (number, %)		43 (52.4)	6 (37.5)	0.41
Familial hypercholesterolemia (number, %)		0 (0.0)	0 (0.0)	---
Previous myocardial infarction (number, %)		6 (7.3)	0 (0.0)	0.59
Severe carotid artery stenosis (number, %)		2 (2.4)	0 (0.0)	1.00
Past or current smoking (number, %)		51 (62.2)	7 (43.8)	0.27
Statin (number, %)		82 (100)	16 (100)	---
Ezetimibe (number, %)		11 (13.4)	2 (12.5)	1.00
Probucol (number, %)		3 (3.7)	3 (18.8)	0.05
Eicosapentaenoic acid (number, %)		67 (81.7)	9 (56.3)	0.045
Antiplatelet agent (number, %)		36 (43.9)	4 (25.0)	0.18
Renin-angiotensin system inhibitor (number, %)		40 (48.8)	6 (37.5)	0.59
β-blocker (number, %)		8 (9.8)	0 (0.0)	0.35
Calcium channel blocker (number, %)		40 (48.8)	8 (50.0)	1.00
Diuretic (number, %)		4 (4.9)	1 (6.3)	1.00
Metformin (number, %)		16 (19.5)	4 (25.0)	0.73
Sodium glucose transporter-2 inhibitor (number, %)		3 (3.7)	0 (0.0)	1.00
Dipeptidyl peptidase 4 inhibitor (number, %)		16 (19.5)	3 (18.8)	1.00
Glucagon-like peptide-1 receptor agonist (number, %)		16 (19.5)	1 (6.3)	0.29
Insulin (number, %)		12 (14.6)	2 (12.5)	1.00
Mean IMT of carotid artery (mm)		1.3 ± 0.2	1.1 ± 0.2	0.004
Maximum IMT of carotid artery (mm)		2.4 ± 0.8	2.2 ± 0.7	0.38
Coronary vulnerable plaque (number)		120	16	
Location of coronary vulnerable plaque	Left anterior descending artery (number, %)	44 (36.7)	10 (62.5)	0.11
	Left circumflex artery (number, %)	23 (19.2)	0 (0.0)	
	Left main trunk (number, %)	5 (4.1)	1 (6.3)	
	Right coronary artery (number, %)	48 (40.0)	5 (31.3)	

Variables are shown as the mean ± standard deviation or number (%). Abbreviation: IMT, intima–media thickness.

**Table 2 jcm-09-03338-t002:** Dose of each statin administered in the evolocumab and control groups.

Statin	Dose (mg/day)	Evolocumab Group (Number, %)	Control Group (Number, %)
Atorvastatin	5	10 (12.2)	3 (18.8)
	10	8 (9.8)	2 (12.5)
	15	1 (1.2)	0 (0.0)
	20	0 (0.0)	0 (0.0)
Pitavastatin	1	1 (1.2)	0 (0.0)
	2	1 (1.2)	0 (0.0)
Pravastatin	5	1 (1.2)	0 (0.0)
	10	6 (7.3)	2 (12.5)
Rosuvastatin	2.5	8 (9.8)	3 (18.8)
	5	23 (28.0)	4 (25.0)
	7.5	5 (6.1)	1 (6.3)
	10	8 (9.8)	0 (0.0)
	15	3 (3.7)	0 (0.0)
	20	7 (8.5)	1 (6.3)

**Table 3 jcm-09-03338-t003:** Simple and multiple linear regression analyses of variables associated with the change in stability (evaluated by minimum CT density) of vulnerable coronary plaques during the administration of evolocumab.

Variables		Simple Linear Regression Analysis		Multivariate Linear Regression Analysis (*p* < 0.10)	
		Standard Coefficient	*p* Value	Standard Coefficient	*p* Value
Age (years)		−0.068	0.47		
Male sex (yes vs. no)		0.020	0.83		
Body mass index (kg/m^2^)		−0.024	0.80		
Systolic blood pressure (mmHg)		−0.050	0.59		
Diastolic blood pressure (mmHg)		0.140	0.14		
HDL-cholesterol (mg/dL)		0.015	0.88		
Change in HDL-cholesterol (mg/dL)		0.142	0.13		
LDL-cholesterol (mg/dL)		0.007	0.94		
Change in LDL-cholesterol (mg/dL)		0.034	0.72		
LDL-cholesterol < 70 mg/dL (yes vs. no)		0.039	0.68		
Log-triglyceride (mg/dL)		−0.022	0.81		
Change in log-triglyceride (mg/dL)		−0.070	0.46		
Log-lipoprotein (a) (mg/dL)		−0.077	0.43		
Change in log-lipoprotein (a) (mg/dL)		−0.019	0.84		
Eicosapentaenoic acid to arachidonic acid ratio		0.012	0.90		
Uric acid (mg/dL)		0.079	0.41		
Change in uric acid (mg/dL)		0.007	0.94		
HbA1c (%)		−0.062	0.52		
Change in HbA1c (%)		0.115	0.22		
eGFR (mL/min/1.73 m^2^)		−0.047	0.62		
Change in eGFR (mL/min/1.73 m^2^)		0.011	0.91		
Log-urine albumin/creatinine ratio (mg/gCr)		−0.035	0.72		
Hypertension (yes vs. no)		−0.083	0.38		
Diabetes mellitus (yes vs. no)		0.073	0.44		
Previous myocardial infarction (yes vs. no)		0.184	0.049	0.193	0.12
Past or current smoking (yes vs. no)		0.129	0.17		
Statin (yes vs. no)		0.000	−−−		
Ezetimibe (yes vs. no)		0.073	0.44		
Probucol (yes vs. no)		0.145	0.12		
Eicosapentaenoic acid (yes vs. no)		−0.003	0.98		
Antiplatelet agent (yes vs. no)		−0.040	0.67		
Renin–angiotensin system inhibitor (yes vs. no)		−0.060	0.53		
β-blocker (yes vs. no)		−0.051	0.59		
Calcium channel blocker (yes vs. no)		−0.086	0.36		
Diuretic (yes vs. no)		0.036	0.70		
Metformin (yes vs. no)		−0.156	0.10		
Sodium glucose transporter-2 inhibitor (yes vs. no)		0.045	0.63		
Dipeptidyl peptidase 4 inhibitor (yes vs. no)		−0.024	0.80		
Glucagon-like peptide-1 receptor agonist (yes vs. no)		−0.129	0.17		
Insulin (yes vs. no)		0.010	0.91		
Location of coronary vulnerable plaque (right coronary artery is the reference group)	Left anterior descending artery	0.166	0.08	0.239	0.06
	Left circumflex artery	0.061	0.52		
	Left main trunk	0.003	0.98		
Mean IMT of carotid artery (mm)		0.179	0.06	0.041	0.74
Maximum IMT of carotid artery (mm)		0.118	0.21		
Baseline minimum CT density of coronary plaque (HU)		−0.188	0.045	−0.029	0.81
Baseline remodeling index of coronary plaque		0.018	0.85		
Baseline percent stenosis at vulnerable coronary plaque site (%)		−0.393	0.002	−0.391	0.002

Abbreviations: CT, computed tomography; eGFR, estimated glomerular filtration rate; HbA1c, serum hemoglobin A1c; HDL, high-density lipoprotein; IMT, intima-media thickness; LDL, low-density lipoprotein; Log, logarithm.

**Table 4 jcm-09-03338-t004:** Simple and multiple linear regression analyses of variables associated with the change in size (evaluated by remodeling index) of vulnerable coronary plaques during the administration of evolocumab.

Variables		Simple Linear Regression Analysis		Multivariate Linear Regression Analysis (*p* < 0.10)	
		Standard Coefficient	*p* Value	Standard Coefficient	*p* Value
Age (years)		0.182	0.05	0.122	0.14
Male sex (yes vs. no)		0.020	0.83		
Body mass index (kg/m^2^)		0.054	0.57		
Systolic blood pressure (mmHg)		0.008	0.93		
Diastolic blood pressure (mmHg)		−0.036	0.70		
HDL-cholesterol (mg/dL)		−0.056	0.55		
Change in HDL-cholesterol (mg/dL)		0.048	0.61		
LDL-cholesterol (mg/dL)		0.118	0.21		
Change in LDL-cholesterol (mg/dL)		−0.287	0.002	−0.143	0.22
LDL-cholesterol < 70 mg/dL (yes vs. no)		−0.186	0.048	0.036	0.75
Log-triglyceride (mg/dL)		0.044	0.65		
Change in log-triglyceride (mg/dL)		0.040	0.67		
Log-lipoprotein (a) (mg/dL)		0.061	0.53		
Change in log-lipoprotein (a) (mg/dL)		0.030	0.76		
Eicosapentaenoic acid to arachidonic acid ratio		−0.117	0.22		
Uric acid (mg/dL)		−0.056	0.55		
Change in uric acid (mg/dL)		0.114	0.23		
HbA1c (%)		0.215	0.023	0.132	0.14
Change in HbA1c (%)		−0.190	0.043	−0.093	0.25
eGFR (mL/min/1.73 m^2^)		0.097	0.31		
Change in eGFR (mL/min/1.73 m^2^)		0.021	0.82		
Log-urine albumin/creatinine ratio (mg/gCr)		0.071	0.47		
Hypertension (yes vs. no)		0.132	0.16		
Diabetes mellitus (yes vs. no)		−0.137	0.15		
Previous myocardial infarction (yes vs. no)		0.085	0.37		
Past or current smoking (yes vs. no)		0.081	0.39		
Statin (yes vs. no)		0.000	−−−		
Ezetimibe (yes vs. no)		−0.154	0.10		
Probucol (yes vs. no)		−0.091	0.34		
Eicosapentaenoic acid (yes vs. no)		0.006	0.95		
Antiplatelet agent (yes vs. no)		0.148	0.12		
Renin–angiotensin system inhibitor (yes vs. no)		0.202	0.031	0.126	0.15
β-blocker (yes vs. no)		−0.007	0.94		
Calcium channel blocker (yes vs. no)		−0.018	0.85		
Diuretic (yes vs. no)		−0.151	0.11		
Metformin (yes vs. no)		0.020	0.84		
Sodium glucose transporter-2 inhibitor (yes vs. no)		0.013	0.89		
Dipeptidyl peptidase 4 inhibitor (yes vs. no)		−0.147	0.12		
Glucagon-like peptide-1 receptor agonist (yes vs. no)		0.041	0.67		
Insulin (yes vs. no)		0.090	0.34		
Location of coronary vulnerable plaque (right coronary artery is the reference group)	Left anterior descending artery	−0.035	0.71		
	Left circumflex artery	0.084	0.37		
	Left main trunk	−0.007	0.94		
Mean IMT of carotid artery (mm)		0.054	0.57		
Maximum IMT of carotid artery (mm)		0.038	0.69		
Baseline minimum CT density of coronary plaque (HU)		−0.017	0.86		
Baseline remodeling index of coronary plaque		−0.421	<0.001	−0.368	<0.001
Baseline percent stenosis at vulnerable coronary plaque site (%)		0.169	0.21		

Abbreviations: CT, computed tomography; eGFR, estimated glomerular filtration rate; HbA1c, serum hemoglobin A1c; HDL, high-density lipoprotein; IMT, intima-media thickness; LDL, low-density lipoprotein; Log, logarithm.

**Table 5 jcm-09-03338-t005:** Laboratory parameters before and 6 months after the baseline in the evolocumab and control groups.

	Evolocumab Group (*n* = 82)			Control Group (*n* = 16)		
	Baseline	6 Month	*p* Value	Baseline	6 Month	*p* Value
Total cholesterol (mg/dL)	149.9 ± 28.5	92.5 ± 22.4	<0.001	174.3 ± 38.3	145.9 ± 22.6	0.021
LDL-cholesterol (mg/dL)	70.4 ± 21.5	19.3 ± 16.0	<0.001	80.5 ± 28.4	68.6 ± 20.5	0.09
HDL-cholesterol (mg/dL)	53.2 ± 12.6	55.7 ± 13.3	<0.001	61.6 ± 12.7	56.8 ± 13.2	0.10
Triglycerides (mg/dL)	111 (81–156)	84 (62–120)	<0.001	101 (69–138)	84 (55–158)	0.25
Lipoprotein(a) (mg/dL)	14 (6–26)	5 (2–14)	<0.001	16 (8–26)	17 (6–29)	0.44
Uric acid (mg/dL)	4.8 ± 1.0	4.7 ± 0.9	0.47	5.2 ± 1.1	5.0 ± 0.9	0.22
HbA1c (%)	6.3 ± 0.8	6.4 ± 0.9	0.09	6.3 ± 1.1	6.2 ± 0.6	0.62
eGFR (mL/min/1.73 m^2^)	68.1 ± 14.8	67.8 ± 14.5	0.79	77.2 ± 13.2	76.0 ± 13.9	0.45
Urine albumin/creatinine ratio (mg/gCr)	9.2 (5.1–28.9)	9.4 (4.9–32.0)	0.66	25.0 (9.0–66.9)	26.4 (7.3–37.9)	0.76
Alanine aminotransferase (IU/L)	21 (15–31)	20 (15–30)	0.95	18 (13–28)	15 (13–19)	0.19
Creatine phosphokinase (IU/L)	116 (80–177)	118 (90–169)	0.18	110 (92–151)	117 (88–234)	0.75

Variables are shown as the mean ± standard deviation or median [interquartile range]. Abbreviations: eGFR, estimated glomerular filtration rate; HbA1c, serum hemoglobin A1c; HDL, high-density lipoprotein; LDL, low-density lipoprotein.

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
