# Peer review of "Effect of Evolocumab on Vulnerable Coronary Plaques: A Serial Coronary Computed Tomography Angiography Study"

_jcm, 2020, doi:10.3390/jcm9103338_

Round 1
Reviewer 1 Report
The authors have adressed my comments.
Author Response
Responses to Reviewer #1
Comment: The authors have addressed my comments.
We thank the reviewer for the comment on our manuscript.
Reviewer 2 Report
The authors have addressed all my concerns; the manuscript has been greatly improved. Congratulations to this nice study.
Author Response
Responses to Reviewer#2
Comment: The authors have addressed all my concerns; the manuscript has been greatly improved. Congratulations to this nice study.
We appreciate the reviewer’s comment for our manuscript.
This manuscript is a resubmission of an earlier submission. The following is a list of the peer review reports and author responses from that submission.
Round 1
Reviewer 1 Report
Manuscript # JCM-879059
Comments for the Authors
Summary
The authors investigated the effects of evolocumab, a fully human monoclonal antibody that inhibits proprotein convertase subtilisin/kexin type 9 (PCSK9), on coronary vulnerable plaques among patients taking statins. Finally, the authors demonstrated that the administration of PCSK9 inhibitor (PCSK9i) was significantly associated not only with improved lipid profile but also with plaque morphology evaluated by coronary computed tomography angiography (CCTA). Although this study is unique, major issues that need clarification are listed below in this manuscript.
Major comments
- A key issue is the study population. In this study, PCSK9i was administered to hypercholesterolemic patients with confirmed cardiovascular risk factors who did not achieve optimal low-density lipoprotein (LDL)-cholesterol goals or exhibited progression of carotid intima–media thickness (IMT) despite statin therapy. As a result, almost two thirds of participants were administered PCSK9i due to the progression of IMT regardless of their LDL level or maximum tolerated dose of a statin, although their mean LDL at baseline was 68.0 mg/dL, under the target for the secondary prevention recommended in the current guideline. Moreover, the dose of statin administered was quite low as shown in Table 2. Such an aggressive usage of PCSK9i seems unusual and challenging strategy that should be tested in a prospective study. Therefore, I am wondering whether the findings from this study can be applied to the general clinical practice. Please clarify the rationale for this advanced approach in accordance with the current guidelines.
- Was PCSK9i effective on the subgroup with low LDL level at baseline? If so, this finding might be interesting and support the aggressive lipid management. Was there a significant association between the decrease of LDL and improved plaque morphology? I believe these additional analyses would provide some insights on how to use PCSK9i for the stabilization of coronary vulnerable plaques.
- The authors stated that the lack of a control group is one of the study limitations. However, I think the authors can recruit patients with vulnerable plaque who did not receive PCSK9i therapy as a control group. The comparison between those who with and without PCSK9i must be of interest as well.
- Since the authors focused on the morphologic change for vulnerable plaques at baseline only, it is unclear whether the favorable effect of PCSK9i on coronary plaques was shown in the whole coronary tree. Is it possible to investigate the morphological change of all plaques? Also, there is a lack of several important plaque characteristics such as plaque volume and percent stenosis. The authors should clarify these detailed data for plaque morphologies evaluated by CCTA.
Minor comments
- Please clarify the definition of the progression of carotid maximum IMT.
- The authors defined 60HU as a cut-off point of low attenuation plaque in CCTA without any references. Is this cut-off value used in the previous articles?
- Patients with previous myocardial infarction were included in this study. How did the authors treat the patients with prior percutaneous coronary intervention or coronary artery bypass grafting? Were these patients excluded?
- In the Discussion (page 13, lines 296-8), the authors described “These results suggest that evolocumab is superior to rosuvastatin in terms of plaque stabilizing and reducing effect on vulnerable coronary plaque.” All patients with PCSK9i received statin therapy in this study; therefore, this is an overstatement.
Reviewer 2 Report
The authors assess the effects of evolocumab, a PCSK9 inhibitor monoclonal antibody, on coronary plaque vulnerability in patients taking statins. The analysis was performed retrospectively using CT analyzed in 276 vulnerable coronary plaques of 144 patients. Evolocumab increased minimum CT density and reduced the remodeling index. The expected decreases in total cholesterol, low-density lipoprotein-cholesterol, triglyceride, and lipoprotein (a) were observed. The major limitation of the study is its retrospective nature. Analogous data have been reported using NIRS and in a previous case report (PMID: 32611955).
Do the authors also have the data on IMT?
Was there a difference in the regression of plaque vulnerability according to sex, baseline lipid levels, location of the stenosis etc? Can the authors provide a multivariable regression analysis on the predictors of plaque stabilization?